# Couple-Based Carrier Screening: How Gene and Variant Considerations Impact Outcomes

**DOI:** 10.3390/genes16060671

**Published:** 2025-05-30

**Authors:** Eric Lee, Kaylee Orton, Edward Kwan, Claire Faga, Trang Le, Ranad Shaheen, Vivek Nair, Simon Cliffe

**Affiliations:** Molecular Genetics Department, Virtus Diagnostics, Suite 4, Level 1, 20-30 Blamey St, Revesby, NSW 2212, Australia

**Keywords:** carrier screening, cystic fibrosis, fragile X syndrome, spinal muscular atrophy, infertility

## Abstract

*Background/Objectives*: The clinical utility of reproductive carrier screening varies based on the genes tested, variant reporting policies, and the screened patient population. This study aims to evaluate the outcomes of carrier screening among reproductive couples undergoing testing in a routine clinical setting. *Methods*: A total of 1595 couples, primarily referred by reproductive endocrinology and infertility specialists, underwent couple-based carrier screening across 390 genes. Carrier states were assessed on a couple basis and reported only if a couple were at risk of having affected offspring. At-risk conditions were classified by severity, as well as their likelihood of clinical impact based on the specific variants detected in each at-risk couple. Secondary findings with potential personal utility were also evaluated. *Results*: Among the screened couples, 4.2% were at risk of having a child with a genetic condition. When limited to high-clinical-impact results, the at-risk couple rate decreased to 1.0%, with 44% of these cases involving *CFTR*, *SMN1*, or *FMR1*. Secondary findings were identified in 1.7% of individuals. *Conclusions*: Carrier screening for only *CFTR*, *SMN1*, and *FMR1* will miss more than half of at-risk couples, underscoring the importance of broader carrier screening. Specific variants and their combinations can influence the predicted clinical impact of at-risk conditions, marking a key advantage of couple-based reporting. Secondary findings were common, highlighting the importance of discussing these potential findings during pre-test counselling.

## 1. Introduction

Carrier screening identifies couples at risk of having a child affected by autosomal recessive or X-linked genetic conditions. Many at-risk couples plan or pursue interventions to avoid the birth of an affected child, such as prenatal diagnostic testing, in vitro fertilisation with preimplantation genetic testing (PGT), or the use of donor gametes [1,2].

Since 2023, government funding in Australia has covered carrier screening for three common conditions: cystic fibrosis, spinal muscular atrophy, and fragile X syndrome. However, these conditions represent a fraction of known autosomal recessive and X-linked conditions, leaving a significant gap in identifying risks for less common yet serious conditions. Broadening carrier screening to include a wider array of conditions is more costly but offers a more comprehensive assessment and is more inclusive of diverse ancestries [3]. This gap between limited and broader carrier screening has been recently investigated in Australia, where a nationwide couple-based carrier screening program using a panel of 1281 genes was undertaken to evaluate its feasibility, acceptability, and outcomes (‘Mackenzie’s Mission’) [2].

One key measure of carrier screening outcomes is the proportion of at-risk couples (ARC) identified. However, reported ARC rates vary significantly between studies (from 0.2% to 11.8%), due to differences in patient populations, panel sizes, inclusion/exclusion of certain common conditions, types of variants detected, and the method of determining coupling (real vs. simulated) [2,4,5,6,7,8,9,10,11,12,13,14].

The selection of genes and conditions for screening is a major source of variability in assessing the effectiveness of carrier screening. Numerous criteria have been proposed for choosing genetic conditions for screening, such as well-defined phenotypes, early-onset disease, shortened lifespan, severe physical or intellectual disability, disabling malformations, significant impact on quality of life, need for medical or surgical interventions, and high population carrier frequencies [3,15,16,17]. Despite such selection criteria, variability in the number and types of genes screened persists among carrier screening providers and in research studies [18].

Another factor contributing to variability in reported screening outcomes is whether or not disease severity is considered when determining the ARC rate. Disease severity can vary not only between conditions but also within the same condition. Some variants in a gene may consistently cause severe phenotypes, while others in the same gene may result in milder phenotypes or have their severity influenced by variants in *cis* or *trans* [19]. This is an important consideration, as predicted disease severity has been shown to influence reproductive decision making, with at-risk couples more likely to take steps to avoid conditions classified as severe compared to those considered mild [1,2,13,20,21].

Carrier screening can also uncover secondary findings beyond its primary purpose of identifying reproductive risk. These include heterozygous variants in genes associated with dominant, actionable conditions, for which there exist well-established guidelines for reporting as secondary findings. However, there is limited guidance on managing other findings, such as potentially biallelic variants indicating unrecognised recessive conditions or X-linked carrier states that may have health implications for the individual. As a result, approaches to consent, disclosure, and follow-up for such secondary findings in the context of carrier screening remain inconsistent and largely unexplored.

These complexities underscore the challenges in accurately evaluating the benefits of broader carrier screening. This study aims to provide a comprehensive evaluation of broader carrier screening outcomes, addressing key limitations of prior studies by assessing the prevalence of at-risk couples in a routine clinical setting, the influence of variant-specific phenotype considerations, and the frequency and clinical relevance of secondary findings.

## 2. Materials and Methods

Study participants were all couples referred to Virtus Diagnostics for couple-based carrier screening between 2023 and 2024, with all tests performed in the context of routine patient care. Virtus Diagnostics provides genetic testing services for adult patients, with most test referrals received from reproductive endocrinology and infertility (REI) specialists. Screening was offered to couples planning for pregnancy or in the early stages of pregnancy. All participating couples consisted of one male and one female individual presenting as reproductive partners.

Participant samples were obtained either from peripheral blood stored in EDTA or from saliva collected using the Oragene OCR-100 Saliva Kit (DNA Genotek, Ottawa, ON, Canada). Genomic DNA extraction was performed using the QIAsymphony DNA Mini Kit on the QIAsymphony SP Instrument (Qiagen, Venlo, The Netherlands).

Sequencing libraries were prepared using the Ion AmpliSeq CarrierSeq Expanded Carrier Screening (ECS) Panel. Template preparation, enrichment, and chip loading were performed on the Ion Chef Instrument, followed by sequencing on the Ion GeneStudio S5 System using Ion 540 Chips (Thermo Fisher Scientific, Waltham, MA, USA). Variant calling and annotation were performed with Ion Reporter software and Carrier Reporter software (Igentify, Caesarea, Israel). Tertiary analysis was conducted using either in-house algorithms or Franklin software (Genoox, Palo Alto, CA, USA). The scope of screened genes was constrained by the 420 genes available in the CarrierSeq panel. After excluding genes associated solely with mild conditions, the final panel used in this study comprised 390 genes, including 361 autosomal and 29 X-linked genes (Appendix A). *FMR1* CGG repeat testing was conducted using the CarrierMax FMR1 Reagent Kit, with PCR amplification products analysed on the Applied Biosystems 3500 XL Genetic Analyer (Thermo Fisher Scientific, Waltham, MA, USA). AGG interrupt testing was not performed.

Clinical validation of the sequencing panel demonstrated sensitivity in detecting *SMN1* exon 7 and *HBA1*/*HBA2* copy number losses, multi-exon copy number variants in other panel genes, and variants in genes associated with pseudogenes (e.g., *CYP21A2*). Validation data confirmed that the assay’s sensitivity for *SMN1* and *HBA1/HBA2* losses was comparable to orthogonal techniques such as multiplex ligation-dependent probe amplification (MLPA); however, sensitivity for multi-exon copy number variants in other genes and variants in genes with pseudogenes could not be quantified due to the small number of tested positive controls. Furthermore, the specificity for all these variant types was low. Consequently, all such reportable findings were confirmed by orthogonal methods, including MLPA, long-range PCR, and Sanger sequencing. Individuals heterozygous for *SMN1* c.*3+80T>G (NM_000344.3) were reported as potential 2+0 carriers if detected alongside two copies of *SMN1* exon 7.

Variant classification was performed according to ACMG (American College of Medical Genetics and Genomics) recommendations or gene-specific ClinGen Variant Curation Expert Panel Protocols [22]. All reported variants, whether identified as carrier states or secondary findings, were classified by the laboratory as pathogenic or likely pathogenic. Variants of uncertain significance were not reported. Carrier states were reported on a couple-basis in a single laboratory report, meaning both partners had to be carriers of pathogenic/likely pathogenic variants in the same autosomal gene for a carrier state in that gene to be reported. If only one partner was found to carry a pathogenic or likely pathogenic variant in a given gene, the carrier state for that gene was not reported. As a result, carrier frequencies for the screened genes are not available for this study group. Exceptions included variants in ‘core’ genes (*CFTR*, *SMN1*, *FMR1*, *HBA1*, *HBA2*, *HBB*), where individual carrier states were reported for both tested partners; *FMR1* repeat alleles and variants in other X-linked genes, which were analysed and reported on an individual basis in female partners only; and secondary findings with potential health implications for individuals, restricted by laboratory policy to heterozygous variants in *LDLR*, *FH*, *ATM*, or *TTN*, or potentially biallelic variants in other genes. The core genes were selected either because Australian government reimbursement for testing is contingent on reporting carrier status in these genes (*CFTR*, *SMN1*, *FMR1*), or due to their known high carrier prevalence in multiple global populations (*HBA1*, *HBA2*, *HBB*).

Gene-specific variant classification and reporting policies included the following. For *SERPINA1*, couples were only reported as at risk if both partners were PI*Z carriers; for *CFTR*, only pathogenic or likely pathogenic variants with at least one associated case of cystic fibrosis in the literature were reported as carrier states; and for *TTN*, only truncating variants in exons expressed in the N2B and N2BA transcripts, with a percent spliced in (PSI) figure of >90% according to cardiodb.org, were reported as secondary findings.

In order to reduce ascertainment bias, individuals already known to be carriers, or identified as at risk prior to testing, were included in the overall study group but excluded from the calculation of carrier frequencies and at-risk couples. Couples in which both partners were carriers but whose offspring were only at risk for non-classic congenital adrenal hyperplasia or alpha-thalassemia trait were also excluded from these calculations. Variants associated exclusively with biochemical phenotypes and without known clinical consequences were not reported.

The severity of at-risk conditions was classified based on the criteria established by Lazarin et al. [17]. For conditions with a known heterogeneous phenotypic spectrum, the most severe untreated disease presentation was used as the reference point for assessment. Each condition was categorised as either “severe/profound” or “mild/moderate” based on this framework. Each couple’s risk status was then further classified as either “high” or “low/medium” clinical impact, considering the specific variants identified in the couple. All findings involving mild/moderate conditions were automatically classified as low/medium clinical impact. In contrast, findings related to severe/profound conditions were individually reviewed by a genetic pathologist, who assessed clinical impact based on factors such as penetrance, variable expressivity, and published evidence regarding specific genotype combinations in the literature and online databases.

## 3. Results

This study included 3190 individuals and 1595 couples, with a mean age of 34.6 years for women and 36.6 years for men. The majority of participants were from the state of New South Wales (72%), with smaller proportions from Victoria (11%), Tasmania (8%), Queensland (8%), and other states. Most couples (96%) were referred by REI specialists, while the remainder were referred by clinical geneticists, obstetrician/gynaecologists, or primary care providers.

Within the core genes (*CFTR*, *SMN1*, *FMR1*, *HBA1*, *HBA2*, *HBB*), 338 participants were carriers in one or more of these genes. Specifically, 95 were *CFTR* carriers (1 in 34 participants), 75 were *SMN1* one-copy carriers (1 in 43 participants), 15 were *FMR1* premutation carriers (1 in 106 female participants), 140 were alpha-thalassemia carriers (1 in 23 participants), and 34 were beta-thalassemia carriers (1 in 94 participants). Of the alpha-thalassemia carriers, 107 (76%) had a one-gene deletion, 18 (13%) were heterozygous for a two-gene deletion, 6 (4%) were homozygous for a one-gene deletion, and 9 (6%) were carriers of a single-nucleotide variant. Among *FMR1* carriers, 11 (73%) had 55–64 repeats, and 4 (27%) had 65 or more repeats. Two couples were identified as being at risk of having offspring with cystic fibrosis, and one couple was at risk of having offspring with spinal muscular atrophy. No couples were found to be at risk for Hb H disease, Hb Bart syndrome, beta-thalassemia, or a haemoglobinopathy.

Among the 1595 couples screened, 67 couples (4.2% of 1595) were identified as being at risk of having a child with an autosomal recessive (AR) or X-linked (XL) condition (Figure 1). This figure excludes eight couples who were already known to be carriers of the same condition at the time of test request. Of these 67 couples, 41 (2.6% of 1595) were at risk for a condition of potentially severe or profound severity, while the remaining cases involved mild or moderate-severity conditions. Within this group, 16 couples (1.0%, 95% CI 0.6–1.6%) received high-clinical-impact results, as previously defined (Appendix A). Overall, core genes accounted for 18 at-risk couples (27% of all at-risk couples) but made up 7 (44%) of high-clinical-impact results.

The gene panel used for screening included 76% of AR and 63% of XL genes in the ACMG-recommended tier 1, 2, and 3 gene lists [3]. For 61 of the 67 at-risk couples (91%), the at-risk gene was included in the ACMG-recommended gene lists. In the remaining six at-risk couples, the gene was not on these lists and involved *ACSF3*, *MYO15A*, *NPHS2*, *SERPINA1*, *SBDS*, and *GJB1* (X-linked). When considering only high-clinical-impact results, the at-risk gene was included in the ACMG gene list for all but one at-risk couple, who were at risk of having offspring with *SBDS*-associated Shwachman–Diamond syndrome.

There were 55 participants (1.7% of 3190, or 1 in 58) with secondary findings reported, including 24 males and 31 females. Of these, 38 (69% of 55) were related to autosomal dominant phenotypes. Variants were most commonly identified in *TTN* (*n* = 18), *LDLR* (*n* = 12), and *ATM* (*n* = 8). Additionally, 17 participants (31% of 55) harboured homozygous or potentially compound heterozygous variants in genes associated with autosomal recessive phenotypes, including *GJB2* (*n* = 4), *SERPINA1* (*n* = 3), *SLC12A3* (*n* = 2), and *CFTR* (*n* = 2) (Appendix A). None of these participants had reproductive partners who were carriers in the same gene or who had their own secondary findings. Four male individuals who were found to have two variants in *CYP21A2* and whose genotypes suggested a potential diagnosis of non-classic CAH (characterised in males by mild signs of hyperandrogenism) were excluded from the secondary findings group.

## 4. Discussion

Couple-based carrier screening across 390 genes identified at-risk couples in 4.2% of couples tested. When limited to high-clinical-impact results, this proportion decreased to 1.0%. Among high-clinical-impact results, 44% were attributable to just three genes: *CFTR*, *SMN1*, and *FMR1*. Secondary findings unrelated to reproductive risk were identified in 1 out of every 58 individuals tested.

### 4.1. Screening for Only Three Genes Will Miss More than Half of At-Risk Couples

*CFTR*, *SMN1*, and *FMR1* accounted for 44% of at-risk couples with high-clinical-impact results, meaning that 56% of at-risk couples with high-clinical-impact results would have gone undetected if screening had been limited to these genes alone, as is currently funded by the Australian Medicare system. These findings are consistent with other studies that have examined the proportion of at-risk couples for these three genes within larger gene panels. In Mackenzie’s Mission, these three genes accounted for 33% of couples at risk for severe or profound conditions [2]. Similarly, in a study using a 114-gene panel, they accounted for 46% of at-risk couples [7]. These findings reinforce that while *CFTR*, *SMN1*, and *FMR1* contribute significantly to at-risk couple identification, limiting screening to these genes alone would miss the majority of high-clinical-impact results.

Broader carrier screening therefore provides a more comprehensive assessment of recessive condition risks than the three-gene screen currently funded in Australia. However, the financial gap between the publicly funded option and broader screening may create barriers to access, disproportionately affecting disadvantaged socioeconomic groups. This could result in a return to historical inequities in reproductive screening [25] despite government funding intended to reduce such disparities.

A total of 1 in 106 female individuals were found to be *FMR1* premutation carriers. Two-thirds of *FMR1* repeats detected were in the 55–64 range, where the risk of expansion to a full mutation is 5% or less, regardless of AGG interrupt status [23]. This frequency is higher than that previously reported in US and Australian populations, which ranged from 1 in 151 to 1 in 330 women [2,8,26,27,28]. This finding aligns with the study’s predominantly REI specialist-referred population, where enrichment of women with *FMR1*-associated premature ovarian insufficiency is expected. These results suggest that using a 65-repeat threshold to stratify individuals by expansion risk in offspring would classify most carriers as low-risk, and such an approach could be implemented in settings with limited access to AGG interrupt testing.

### 4.2. Variant Combinations Affect the Predicted Clinical Impact of At-Risk Conditions

The proportion of at-risk couples varied significantly depending on which conditions and variants were included in the reported percentage. When all reported conditions and variants were considered, 4.2% of couples were classified as at risk. However, when restricted to disorders of potentially severe or profound severity, this proportion decreased to 2.6%. Subsequent stratification by likelihood of clinical impact, based on the specific variants identified within each couple, further reduced the at-risk couple rate to 1.0% (Figure 1). This stratified approach highlights how post-test considerations, such as assessing the clinical impact of specific variant combinations, can substantially influence the proportion of couples considered as at-risk in carrier screening programs.

The at-risk couple rate of 1.0% observed in our study is consistent with findings from Mackenzie’s Mission, which reported a similar rate of 1.2% for couples at risk of having children affected by severe or profound conditions [2]. The similarity in at-risk rates, despite Mackenzie’s Mission screening a substantially larger number of genes (1281 genes), may reflect several counterbalancing factors, including differences in study populations, assay capabilities, and approaches to defining condition severity. This underscores the complexity of assessing carrier screening outcomes at a population level, where nuanced variables beyond the number of genes screened play an important role in determining clinical yield.

The findings of this study challenge the notion that severity is primarily curated at the gene panel selection stage, emphasising instead that accurate risk assessment must also account for specific variant combinations within each couple. The ability to do so marks a key advantage of couple-based carrier screening reports. A simplistic gene-only approach, which assumes that all couples carrying pathogenic variants in the same gene are equally at risk of a clinically significant condition in their offspring, may overestimate risk and overlook critical nuances that could inform reproductive decision making. Studies have shown that couples often make decisions based on the ‘worst-case scenario’ in terms of disease severity, and couples facing risk of high-severity conditions are more likely to pursue reproductive interventions [1,2,13,20,21].

These findings have important implications for policy makers. As carrier screening expands in scope and increasingly moves toward public funding or reimbursement, the reported rate of at-risk couples will become a key metric for assessing clinical utility and cost-effectiveness. It is therefore important that these figures reflect clinically meaningful risk, rather than overstating the potential benefits.

This study also reinforces existing calls to integrate variant combination analysis into carrier screening for autosomal recessive conditions [12,18,19,20]. Prior recommendations have even proposed the creation of an international database of autosomal recessive variant combinations and their phenotypic outcomes [19], which could enhance both carrier screening and genomic newborn screening by improving phenotype prediction. However, this type of analysis is currently rarely performed at the laboratory level. Our findings suggest that variant combination analysis, given its impact on clinical interpretation and reproductive planning, should become standard laboratory practice wherever couple-based carrier screen reporting is available.

### 4.3. Secondary Findings Are Common

The percentage of couples in which a secondary finding was identified in at least one partner (1.7%) was higher than the percentage of couples with high-clinical-impact reproductive findings (1.0%). Approximately two-thirds of these findings were related to dominant phenotypes, while one-third involved recessive phenotypes. The frequency of secondary findings would be even higher if X-linked variants were included. However, these were excluded from consideration due to their often unpredictable clinical consequences in females, largely attributable to variable X-inactivation and resulting differences in penetrance and expressivity (Appendix A).

Autosomal dominant findings were limited to variants in *LDLR*, *TTN*, and *ATM*. Although *LDLR* and *TTN* genes are included in ACMG secondary finding guidelines, the *ATM* gene is not included and is classified as a moderate-risk allele for breast cancer. However, both the NCCN (National Comprehensive Cancer Network) and eviQ recommend annual mammography screening from age 40 for female *ATM* carriers, with modifications to surveillance based on individual risk assessments using validated models such as CanRisk [29]. A range of potentially biallelic variants were also identified (Appendix A).

This aligns with other studies, which have reported findings with health implications in 1 to 2.3% of screened individuals [7,30,31]. These results show that secondary findings are relatively common when large gene panels are used and underscore the importance of discussing this possibility during pre-test counselling. Further research is needed to evaluate the clinical utility, psychological impact, and patient and provider acceptance of disclosing secondary findings in the context of carrier screening.

In parallel, growing evidence suggests that large-scale genomic screening can address gaps missed by clinical indication-based testing [32,33]. Therefore, it may be opportune for patients presenting for carrier screening to be screened for other genetic risks with personal utility at the same time, provided appropriate pre-test counselling is conducted.

This study has several strengths, including the analysis of a group of reproductive couples in a routine clinical setting and the comprehensive detection of genetic variants, including copy number variants and variants in technically challenging genes. Moreover, the study provides a detailed outline of detected variants and expected offspring phenotypes for each couple. These strengths help address limitations seen in prior studies, including a reliance on simulated couple pairings, incomplete or contingent partner testing, reduced sensitivity for copy number or technically challenging variants, and the inclusion of mild conditions in at-risk couple calculations [2,4,5,6,7,8,9,10,11,12,13].

This study has several limitations. First, the study group was predominantly referred by REI specialists and consisted of couples undergoing infertility investigations, which could result in a higher prevalence of carrier states for certain conditions. This is reflected in the increased frequency of *FMR1* carrier states compared to previous reports in the literature. However, few of the other at-risk conditions or secondary findings were directly linked to infertility or increased miscarriage risk. Second, the lack of self-reported or genotype-based ancestry information among study participants limits our ability to assess representation across the broader population. Third, the classification of clinical impact, which informed the adjusted at-risk couple rate, was performed by a single pathologist. While this introduces some subjectivity, the classifications were based on current medical knowledge, and the key sources used for these evaluations have been made available in the Appendix A. Fourth, this laboratory-based study did not assess clinical outcomes subsequent to the reporting of these results, such as reproductive decision making or phenotypic confirmation of secondary findings. Nonetheless, the genotype-based conclusions remain valid based on current evidence and provide valuable insights into carrier screening outcomes.

## 5. Conclusions

This study highlights the value of couple-based broader carrier screening. Screening for only *CFTR*, *SMN1*, and *FMR1* would miss more than half of at-risk couples. The variation in at-risk couple rates, once condition severity and likelihood of clinical impact are considered, highlights the importance of gene panel selection as well as careful assessment of individual variants and their combinations within each reproductive couple. Secondary findings are common, mandating their discussion during pre-test counselling.

## Figures and Tables

**Figure 1 genes-16-00671-f001:**
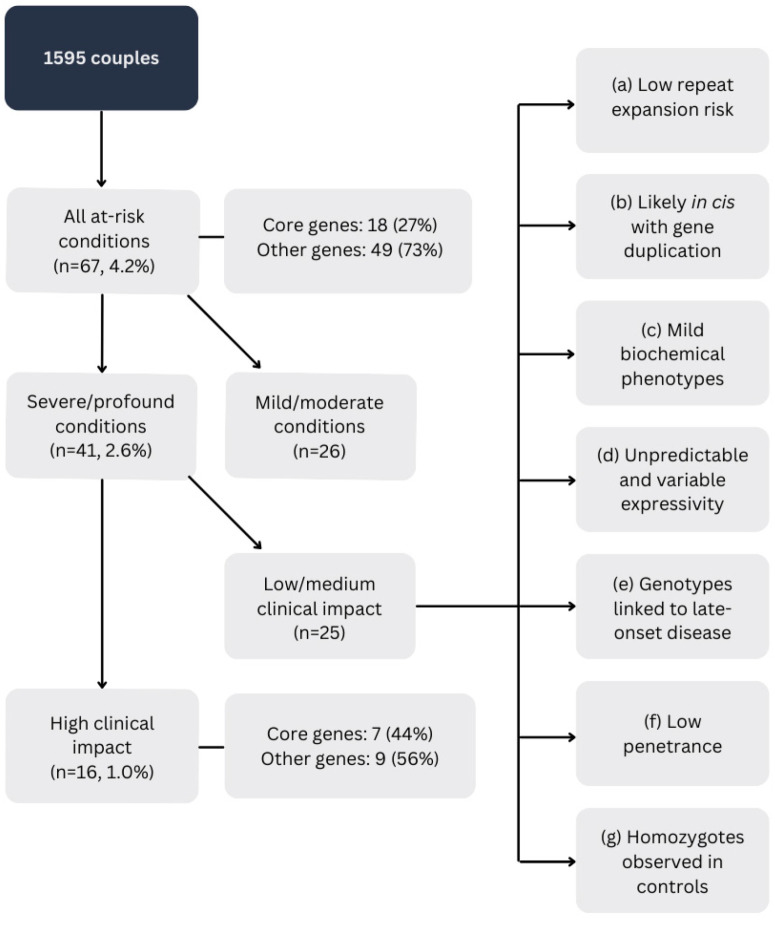
At-risk couples. Core genes: *CFTR, SMN1, FMR1, HBA1/HBA2, HBB*; (a) example: *FMR1* CGG repeats <65 have ≤5% risk of expansion to full mutation, even in the absence of AGG interruptions [23]; (b) example: at least one partner is a carrier of the *CYP21A2* p.(Gln319*) variant and a whole-gene duplication, which are found *in cis* in 84% of carriers [24]; (c) examples: genotypes linked to *BTD*-related partial biotinidase deficiency or *PAH*-related mild hyperphenylalanaemia; (d) example: *POLG*-related disorders; (e) example: *GAA*-related Pompe disease and *NPHS2*-related steroid-resistant nephrotic syndrome; (f) example: *SERPINA1* PI*ZZ homozygotes have a 2% risk of severe liver disease in childhood (GeneReviews—Alpha-1 Antitrypsin Deficiency); (g) examples: homozygotes for *AGXT* p.(Pro11Arg) and *ACSF3* p.(Arg558Trp) have been observed in controls populations.

## Data Availability

The original contributions presented in this study are included in the article/Appendix A. Further inquiries can be directed to the corresponding author.

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
