# Peer review of "Couple-Based Carrier Screening: How Gene and Variant Considerations Impact Outcomes"

_genes, 2025, doi:10.3390/genes16060671_

Round 1

Reviewer 1 Report

Comments and Suggestions for Authors

The manuscript entitled “Couple-Based Carrier Screening: How Gene and Variant Considerations Impact Outcomes” primarily seeks to assess the results of carrier screening among a cohort of actual reproductive couples, particularly emphasizing the significance of comprehensive genetic screening. The study was well executed and scientifically presented. However, the following concerns need to be taken care.

Comments:

Is there any available information regarding the family history of the participants identified as having significant genetic risks? Understanding the family history of the patients could prove beneficial in the meticulous selection of the gene panel for screening purposes.

How did the authors choose a panel of 390 genes for their screening process? Is this panel already available for the Australian population?

The couples included in the study were undergoing infertility investigations. Did the authors identify any genetic factors associated with their infertility challenges?

What is the current prevalence of cystic fibrosis, spinal muscular atrophy, and fragile X syndrome in the study population?

Emphasize the difficulties associated with the accessibility of comprehensive genetic testing.

Author Response

  • Comment 1: Is there any available information regarding the family history of the participants identified as having significant genetic risks? Understanding the family history of the patients could prove beneficial in the meticulous selection of the gene panel for screening purposes.

Thank you for this question. The study participants underwent testing as part of routine clinical care, and in many cases, limited or no family history or clinical information was available. We fully agree that when a family history is known, it is essential to ensure that the gene panel includes relevant genes and that known familial variants are detectable by the chosen assay.

  • Comment 2: How did the authors choose a panel of 390 genes for their screening process? Is this panel already available for the Australian population?

Thank you for this question. As noted on page 3, lines 94–97: “The scope of screened genes was constrained by the 420 genes included in the CarrierSeq panel. After excluding genes associated solely with mild conditions, the final panel used in this study comprised 390 genes, including 361 autosomal and 29 X-linked genes.”

This panel was developed for and made available to the Australian population as part of clinical testing.

  • Comment 3: The couples included in the study were undergoing infertility investigations. Did the authors identify any genetic factors associated with their infertility challenges?

Thank you for this question. As noted on page 7, lines 298–303: “This study has several limitations. First, the study group was predominantly referred by REI specialists and consisted of couples undergoing infertility investigations, which could result in a higher prevalence of carrier states for certain conditions. This is reflected in the increased frequency of FMR1 carrier states compared to previous reports in the literature. However, few of the other at-risk conditions or secondary findings were directly linked to infertility or increased miscarriage risk.”

We did not identify specific genetic factors contributing to infertility in these couples. However, this assessment was limited by the lack of detailed clinical information on the nature of each couple’s fertility challenges.

  • Comment 4: What is the current prevalence of cystic fibrosis, spinal muscular atrophy, and fragile X syndrome in the study population?

Thank you for this question. The prevalence of CFTR-related conditions and FMR1 premutations in our study population is likely higher than in the general population, as most participants were referred by REI specialists. These genes are known to be associated with male and female fertility issues, respectively. Note that individuals with known CFTR-related disorders or biallelic CFTR variants were excluded when calculating carrier frequencies and at-risk couple rates. In contrast, SMN1 (spinal muscular atrophy) carrier prevalence is not expected to differ significantly from the general population.

Despite these referral patterns, the contribution of these three genes to at-risk couple identification was similar to that reported in non-fertility-enriched cohorts. As noted on page 5, line 213: “CFTR, SMN1, and FMR1 accounted for 44% of at-risk couples with high-impact results, meaning that 56% of at-risk couples with high-impact results would have gone undetected if screening had been limited to these genes alone, as is currently funded by the Australian Medicare system. These findings are consistent with other studies that have examined the proportion of at-risk couples for these three genes within larger gene panels. In MacKenzie’s Mission, these three genes accounted for 33% of couples at risk for severe or profound conditions [2] Similarly, in a study using a 114-gene panel, they accounted for 46% of at-risk couples [7]”

  • Comment 5: Emphasize the difficulties associated with the accessibility of comprehensive genetic testing.

Thank you for this excellent suggestion. We had previously addressed the issue of accessibility on page 6, paragraph 2, line 223: “Broader carrier screening therefore provides a more comprehensive assessment of recessive condition risks than the three-gene screen currently funded in Australia. However, the financial gap between the publicly funded option and broader screening may create barriers to access, disproportionately affecting disadvantaged socioeconomic groups. This could result in a return to historical inequities in reproductive screening [25], despite government funding intended to reduce such disparities.”

We agree that barriers to access extend beyond financial considerations, including limited provider awareness and unequal access to pre- and post-test counselling. However, as our study did not specifically investigate these factors, we have chosen not to expand further on these points to remain within the bounds of the data presented.

Reviewer 2 Report

Comments and Suggestions for Authors

Lee and colleagues present a well written description of their experience in reproductive carrier screening for an expanded list of genetic disorders. In particular, they provide a nuanced explanation of how evaluating findings at a gene and variant level impact findings. Their report is concise, succinct, and should help improve clinical care. More detail is needed to place their study in the context of other findings, particular with regards to MacKenzie’s Mission. I elaborate on this point and others below. I think these points can be easily addressed. I appreciate the chance to review this work.

  1. Authors note that broadening carrier screening would benefit families from broader ancestries (Lines 38–40). This is an important point, which would benefit from more detail. Authors should provide more detail regarding which autosomal recessive and X-linked conditions are prevalent if different ancestral groups.
  2. In the introduction, authors briefly refer to MacKenzie’s Mission (Ref 2), describing a study of expanded carrier screening in Australia. Given the similarity of MacKenzie’s Mission to this project, further detail should be provided in the introduction. Authors should explicitly state the outcomes from MacKenzie’s Mission: what portion of couples were found to be at-risk for having a child with a genetic disorder? How often were “high-impact” results present? How often were secondary findings present? Furthermore, authors should explain what gaps, if any, from the MacKenzie’s Mission are being addressed in this manuscript. Authors state that this project addresses “key limitations of prior studies by assessing the prevalence of at-risk couples in real-world reproductive pairings, the influence of variant-specific phenotype considerations, and the frequency and clinical relevance of secondary findings.” It is unclear which of these factors were not addressed in the MacKenzie’s Mission. Additionally, authors should explain the degree to which their study tests the replicability of the findings of MacKenzie’s Mission. Simply replicating findings would still be a worthwhile contribution.
  3. Further detail should be given regarding the 390 genes of the panel. How were these genes selected? Why was this panel selected? How do these genes differ from the 1281 screened in MacKenzie’s Mission (Ref 2).
  4. Authors state that couples known to be carriers prior to testing were included in overall group, but excluded from calculation of carrier frequency and at-risk status (Lines 134–136). It would be helpful if authors include how often this was the case in results.
  5. Authors state that one of the potential benefits of expanded carrier screening is screening for genetic diseases that affect “diverse ancestries.” This is an important consideration, especially as this was an evaluation of screening in a “real-world” setting. To support this, authors should report self-identified ancestry of study participants. Furthermore, it would be informative to report carrier status findings stratified by ancestry.
  6. Authors note similarity of their findings to those of MacKenzie’s Mission in the discussion. It would serve this comparison if authors could report the portion of the 390 genes in their screening that were also included in MacKezie’s Mission.
  7. More attention should be given in the discussion to the generalizability of findings. Sociodemographic data (ancestry, private/public insurance status, etc.) in results would facilitate this.
  8. It would be informative if the authors could compare their findings to results from newborn screening, particularly the recently published Guardian Study. I’m particularly curious to know if the expected rates of genetic disorders in this cohort (had they not undergone screening) would have approached the rates of genetic disorders observed in Guardian. This would also help explain generalizability.

Author Response

  • Comment 1: Authors note that broadening carrier screening would benefit families from broader ancestries (Lines 38–40). This is an important point, which would benefit from more detail. Authors should provide more detail regarding which autosomal recessive and X-linked conditions are prevalent if different ancestral groups.

Thank you for this valuable comment. We agree that broadening carrier screening has particular benefits for individuals from diverse ancestral backgrounds. However, due to the absence of detailed ancestry data in our study cohort, we are limited in our ability to provide a meaningful analysis of condition prevalence by ancestry. We have therefore chosen not to expand further on this complex topic within the Introduction, as it is not subsequently addressed by the data presented in our study. We do, however, refer readers to the ACMG carrier screening guidelines, which offer a comprehensive discussion of ancestry-related considerations.

  • Comment 2: In the introduction, authors briefly refer to MacKenzie’s Mission (Ref 2), describing a study of expanded carrier screening in Australia. Given the similarity of MacKenzie’s Mission to this project, further detail should be provided in the introduction. Authors should explicitly state the outcomes from MacKenzie’s Mission: what portion of couples were found to be at-risk for having a child with a genetic disorder? How often were “high-impact” results present? How often were secondary findings present? Furthermore, authors should explain what gaps, if any, from the MacKenzie’s Mission are being addressed in this manuscript. Authors state that this project addresses “key limitations of prior studies by assessing the prevalence of at-risk couples in real-world reproductive pairings, the influence of variant-specific phenotype considerations, and the frequency and clinical relevance of secondary findings.” It is unclear which of these factors were not addressed in the MacKenzie’s Mission. Additionally, authors should explain the degree to which their study tests the replicability of the findings of MacKenzie’s Mission. Simply replicating findings would still be a worthwhile contribution.

Thank you for this comprehensive and constructive comment. We would like to clarify that our statement regarding the “key limitations of prior studies” in the Introduction refers not only to Mackenzie’s Mission but also to a broader body of literature on carrier screening, as referenced. The following paragraph in introduction outlines some of these limitations (page 2, lines 45-48): “However, reported ARC rates vary significantly between studies (from 0.2% to 11.8%), due to differences in patient populations, panel sizes, inclusion/exclusion of certain common conditions, types of variants detected, and method of determining coupling (real vs. simu-lated) [2,4–14]”

We acknowledge the importance of providing more detailed context on Mackenzie’s Mission, given its immediate relevance to this study. While the Discussion section already compares our findings with those of Mackenzie’s Mission, we agree that further elaboration would strengthen the manuscript. We have therefore expanded the relevant section of the Discussion (Page 6, from line 263) to more clearly delineate the similarities and differences between the two studies:

“The at-risk couple rate of 1.0% observed in our study is consistent with findings from Mackenzie’s Mission, which reported a similar rate of 1.2% for couples at risk of having children affected by severe or profound conditions [2] The similarity in at-risk rates, de-spite Mackenzie’s Mission screening a substantially larger number of genes (1281 genes), may reflect several counterbalancing factors, including differences in study populations, assay capabilities, and approaches to defining condition severity. This underscores the complexity of assessing carrier screening outcomes at a population level, where nuanced variables beyond the number of genes screened play a critical role in determining clinical yield.”

Secondary findings of the nature that we describe were specifically excluded from MacKenzie’s Mission, and therefore reference to it in relation to secondary findings has not been made.

  • Comment 3: Further detail should be given regarding the 390 genes of the panel. How were these genes selected? Why was this panel selected? How do these genes differ from the 1281 screened in MacKenzie’s Mission (Ref 2).

Thank you for this question. As noted on page 3, lines 94–97: “The scope of screened genes was constrained by the 420 genes included in the CarrierSeq panel. After excluding genes associated solely with mild conditions, the final panel used in this study comprised 390 genes, including 361 autosomal and 29 X-linked genes.”

We acknowledge that a direct comparison of the gene content between our 390 gene panel and the 1281 genes screened in Mackenzie’s Mission would be of interest. However, this comparison is complicated by technical and methodological differences between the studies. Notably, our panel included detection of copy number variants and variants in several technically challenging genes that were not consistently detectable across all laboratories participating in Mackenzie’s Mission (E. Kirk, personal communication). As such, panel size alone does not fully reflect the breadth or depth of screening capability.

Furthermore, it is well recognised that expanding gene panels beyond a certain point yields diminishing returns in terms of additional at-risk couple detection. Differences in yield are typically driven by a relatively small subset of genes with high carrier frequencies in the general population. Therefore, while panel composition differs, the clinical yield may be shaped more by these high-impact genes than by the absolute number of genes included.

  • Comment 4: Authors state that couples known to be carriers prior to testing were included in overall group, but excluded from calculation of carrier frequency and at-risk status (Lines 134–136). It would be helpful if authors include how often this was the case in results.

Thank you for this helpful suggestion. We agree that reporting this figure improves transparency. We have now included this detail in the Results section (page 4, lines 182–183): “This figure excludes eight couples who were already known to be carriers of the same condition at the time of test request.”

  • Comment 5: Authors state that one of the potential benefits of expanded carrier screening is screening for genetic diseases that affect “diverse ancestries.” This is an important consideration, especially as this was an evaluation of screening in a “real-world” setting. To support this, authors should report self-identified ancestry of study participants. Furthermore, it would be informative to report carrier status findings stratified by ancestry.

Thank you for this excellent suggestion. As this was a retrospective study using data derived from routine clinical testing, we were limited by the information available at the time of test request. Unfortunately, self-identified ancestry was not collected by the laboratory, and we acknowledge that this limits our ability to directly assess ancestry-related trends in carrier status.

We recognise the importance of understanding the genetic ancestry composition of the cohort and explored an alternative approach using patient surnames, as previously applied in another study by some of the authors (PMID: 39740802). However, surname-based ancestry inference carries several known limitations, and we determined that the data were not sufficiently robust in this instance to include in our analysis.

We agree that future studies would benefit from prospective collection of self-reported ancestry and/or genomic ancestry estimation to better understand the relationship between ancestry and carrier status.

  • Comment 6: Authors note similarity of their findings to those of MacKenzie’s Mission in the discussion. It would serve this comparison if authors could report the portion of the 390 genes in their screening that were also included in MacKezie’s Mission.

Thank you for this helpful suggestion. While we agree that comparing gene panel content can provide useful context, as outlined in our response to Comment 3, a full gene-by-gene comparison is beyond the scope of this study and has limitations in its interpretability.

  • Comment 7: More attention should be given in the discussion to the generalizability of findings. Sociodemographic data (ancestry, private/public insurance status, etc.) in results would facilitate this.

Thank you for this important suggestion. As this study was conducted as a retrospective audit of routine clinical testing, sociodemographic data (such as ancestry and socioeconomic background) were not collected or available to the laboratory.

While we acknowledge that such data can enhance interpretation, we believe their absence does not substantially limit the generalisability of our findings. The rate of at-risk couples for recessive conditions is much more likely to be influenced by genetic ancestry than by socioeconomic status, for example (please see previous response regarding the impact of ancestry).

Nonetheless, we agree that future prospective studies would benefit from collecting sociodemographic variables to better understand access, equity, and potential disparities in carrier screening uptake or outcomes.

  • Comment 8: It would be informative if the authors could compare their findings to results from newborn screening, particularly the recently published Guardian Study. I’m particularly curious to know if the expected rates of genetic disorders in this cohort (had they not undergone screening) would have approached the rates of genetic disorders observed in Guardian. This would also help explain generalizability.

Thank you for this thoughtful suggestion. We have reviewed the results of the recently published Guardian Study (PMID: 39446378). While we are keen to further demonstrate the generalisability of our findings, we believe that direct comparison between our study and the Guardian study may not be feasible for several reasons:

1) Differences in population ancestries: The demographic composition of the screened populations likely differs significantly, particularly with regard to the underrepresentation of Hispanic and Black populations in Australia compared to those in the Guardian study.

2) Variations in screening scope: The Guardian study included screening for conditions such as G6PD deficiency (which accounted for 74% of positive results) and autosomal dominant conditions, which were not part of our screening panel.

3) Statistical power: Both studies may be underpowered for direct comparison, given the rarity of most of the recessive conditions screened; this rarity would introduce a degree of stochastic variation in which specific conditions were identified as screen-positive in each study.

Reviewer 3 Report

Comments and Suggestions for Authors

Authors (all from a diagnostic company) published results they acquired during daily screening work. 

The results they present are neither new nor surprising - in case one checks more infertility related genes onw finds more variants than screening only for three standard genes. 

This finding was the basis to set up the corresponding company.

So this paper is just an advertisement and no scientific work

Minor comment

line 168 _ check the sentence: Within the core genes 338 participants were carriers in one or more of these genes. This makes no sense - it should be stated that authors talk about variants - not genes. 

Author Response

  • Comment 1: The results they present are neither new nor surprising - in case one checks more infertility related genes onw finds more variants than screening only for three standard genes.

Thank you for this feedback. While it is expected that screening more genes will identify more at-risk couples, this study adds value by quantifying the extent of that increase, and showing how the impact varies depending on whether the severity of the predicted phenotype in offspring is taken into account. We believe this nuanced analysis contributes meaningfully to the clinical and policy discussion around expanded reproductive carrier screening.

  • Comment 2: This finding was the basis to set up the corresponding company.

We respectfully disagree. Virtus Diagnostics, where this research was conducted, is a well-established organisation that has provided reproductive genetic testing services for several decades. The company was not created in response to this study, but rather supported the research as part of its ongoing commitment to advancing clinical diagnostics.

  • Comment 3: So this paper is just an advertisement and no scientific work

We respectfully disagree. This comment is not supported by evidence. The paper presents original data and analysis and adheres to scientific and ethical standards. If there are specific concerns about the methodology or findings, we would welcome constructive feedback.

Reviewer 4 Report

Comments and Suggestions for Authors

Overall, this is an interesting manuscript that highlights factors that influence the proportion of at-risk couples identified through carrier screening, depending on the criteria used. The study is focused on the laboratory perspective. This manuscript could be improved by adding clarifications and more detailed descriptions of the study design, data sources, and analyses done as part of the study. 

Specific comments:

Introduction:

  • In the introduction, the authors mention that the disclosure and management of secondary findings in carrier screening are unexplored. Do they expect this to be different than how they are managed in other settings? What are the arguments in favor of adding this to carrier screening? As opposed to exome or genome sequencing that will inevitably sequence the genes targeted for secondary findings, carrier screening for a limited number of genes can be done without sequencing such genes. These are not then really "secondary" findings, they are an add-on test.

Methods

  • In the methods, the participants are briefly described but it is unclear how they were selected and if there were eligibility criteria. Is this the totality of the couples tested in their lab? At the end of the manuscript, it says that consent was waived and ethical review and approval was not required because it is a retrospective analysis of deidentified data obtained for clinical testing purposes. The study design is not clear in the methods. The authors should describe the study design and selection of participants, as well as what analyses were done clinically and what analyses were done for the purpose of the study. For example, were secondary findings reported clinically or assessed for this study?
  • The authors should also describe the data sources if they include information beyond what is available from the laboratory database. 
  • The authors say that they excluded genes associated solely with mild conditions. How was this done? How was it determined that a condition is "mild"? Why were genes for mild conditions on the CarrierSeq panel?
  • The technical details about clinical validation of the panel are well written and add to the validity of this study. 
  • The authors should explain how the "core" genes were selected and why individual carrier status was disclosed for these genes. 
  • The authors should clarify the genes targeted for disclosure as secondary findings and the process that led to this policy. They state that disclosure is restricted to heterozygous variants in four genes and "potentially biallelic variants in other genes". This is markedly different than the ACMG recommendations. Does this represent a national consensus? a local consensus?
  • It is unclear how "high" vs "low/medium" impact are defined.  In the methods, low/medium impact seems to be related to the likelihood of a clinical impact. Is this based on penetrance? Does it take into consideration variable expressivity, type of impact, actionability, or any other factor that could influence clinical impact? The authors should describe their definitions and add references where the definitions could be found, if available. They should also describe how the impact is assessed for each variant/genotype (e.g. who/how many team members assess a variant? how is consensus reached?)
  • Figure 1 should be referenced in the results, not the methods. 
  • The numbers in figure 1 are a little confusing:
    • The boxes that list how many core genes vs other genes are just a description of the box to their left, not an additional step in the process. They should be represented differently. 
    • if including the step "excluding low/moderate impact results", the authors should also include a step described as "excluding mild/moderate conditions"
    • the tables use the terminology "low/medium" impact, whereas the figure uses the term "low/moderate" impact. The same terminology should be used throughout the manuscript. 
    • if including all the different reasons to consider a condition to be "low/medium impact", it would be more interesting to know how many conditions were excluded for each of these reasons. 

Results

  • I suggest moving the sentence "Among FMR1 carriers, the majority (73%) had 55–64 repeats." right after the sentence about alpha-thalassemia carriers, because it describes carriers, before moving to the description of at-risk couples for the core conditions.
  • The results in the paragraph reporting carriers for core genes are confusing because they report absolute numbers, ratios (1 in X) and percentages. The percentages are about carriers of a specific condition, but don't all relate back to the total number of carriers for that condition. For example, only a percentage is given for FMR1 carriers, whereas absolute numbers and percentages are given for alpha-thalassemia carriers. 
  • Supplementary table S1 is not very helpful: some diseases classified as severe/profound are classified as having a low/medium impact and it isn't clear how the severity of the predicted phenotype has led to a downgrade of the impact (e.g. classic congenital adrenal hyperplasia is classified as a "severe/profound disease" but with a "low/medium" impact.
  • About secondary findings, they seem to have included variants of unknown significance and of conflicting classification (table S3). This is at odds with the ACMG guidelines. How do the authors justify reporting such findings?
  • For secondary findings for autosomal recessive phenotypes, the pathogenicity classification of variants is not reported. 

Discussion

  • The authors argue that broader carrier screening  provides a more comprehensive identification of couples at risk for AR or X-linked conditions and that the currently funded public option in Australia may lead to unequal access to broader screening.  They do not address the added costs and resources needed for broader screening, nor the social acceptability of broader screening. 
  • The author rightly acknowledge that their study population is enriched in women with FMR1 carrier status. 
  • Authors discuss the percentage of at-risk couples identified with different strategies. It seems that part of their argument is that the test may target a broader list of genes/conditions but the number of at-risk couples identified can remain manageable if only those with variants in genes associated with severe/profound conditions and predicted high impact are reported. Is that what the authors are arguing? This could be made clearer in the discussion. 
  • Authors should discuss the challenges of predicting clinical impact on the basis of specific variants.
  • The inclusion of secondary findings in the context of carrier screening could be discussed in greater detail, especially if not currently done clinically. 
  • Avoid the formulation "real world couples". Even in research studies, participating couples are real world couples. The emphasis should be on the testing context, not the couple. For example, "this study has several strengths, including the comprehenisve detection of genetic variants in couples testing in real-world settings"

Author Response

  • Comment 1: In the introduction, the authors mention that the disclosure and management of secondary findings in carrier screening are unexplored. Do they expect this to be different than how they are managed in other settings? What are the arguments in favor of adding this to carrier screening? As opposed to exome or genome sequencing that will inevitably sequence the genes targeted for secondary findings, carrier screening for a limited number of genes can be done without sequencing such genes. These are not then really "secondary" findings, they are an add-on test.

Thank you for this thoughtful comment. Our intention was to highlight that while the reporting of secondary findings (particularly for dominant, actionable conditions) is well established in diagnostic settings, there remains limited guidance on managing other categories of findings in the context of carrier screening. These include potentially biallelic variants suggestive of undiagnosed recessive conditions in the individual, and X-linked variants that may have health implications for female carriers.

Importantly, none of the genes in our panel were included solely to identify secondary findings. All genes were selected due to their relevance in autosomal recessive or X-linked disease, consistent with the primary purpose of carrier screening. However, there are several that are associated with both dominant and recessive phenotypes (e.g., LDLR, FH, ATM, TTN) (page 3, line 129). Excluding them entirely would reduce our ability to identify at-risk couples for the recessive conditions associated with these genes.

Thus, we view these findings not as “add-on” or optional results, but as intrinsic to the design and interpretation of a comprehensive carrier screen.

We have amended the paragraph in the Introduction to clarify our intent, on page 2, lines 65-72: “Carrier screening can also uncover secondary findings beyond its primary purpose of identifying reproductive risk. These include heterozygous variants in genes associated with dominant, actionable conditions, for which there exist well-established guidelines for reporting as secondary findings. However, there is limited guidance on managing other findings, such as potentially biallelic variants indicating unrecognised recessive conditions, or X-linked carrier states that may have health implications for the individual. As a result, approaches to disclosure, follow-up, and consent for such secondary findings in the context of carrier screening remain inconsistent and largely unexplored.”

  • Comment 2: In the methods, the participants are briefly described but it is unclear how they were selected and if there were eligibility criteria. Is this the totality of the couples tested in their lab? At the end of the manuscript, it says that consent was waived and ethical review and approval was not required because it is a retrospective analysis of deidentified data obtained for clinical testing purposes. The study design is not clear in the methods. The authors should describe the study design and selection of participants, as well as what analyses were done clinically and what analyses were done for the purpose of the study. For example, were secondary findings reported clinically or assessed for this study?

Thank you for this helpful comment. We clarify that the study cohort included all consecutive couples referred to the laboratory for couple-based carrier screening between 2023 and 2024. There were no specific eligibility criteria beyond referral for testing, and no participants were excluded. All testing was requested by clinicians as part of routine patient care, primarily in the context of preconception or early pregnancy planning.

All analyses and reporting, including any secondary findings, were performed with the intention of providing clinical results to referring healthcare providers. No analyses were conducted solely for research purposes. The data used in this retrospective study were fully de-identified and obtained from clinical records.

We have revised the Materials and Methods section (page 2, from line 80) to clarify the study design, participant selection, and the clinical context in which the testing and reporting took place:

“Study participants were all couples referred to Virtus Diagnostics for couple-based carrier screening between 2023 and 2024, with all tests performed in the context of routine patient care. Virtus Diagnostics provides genetic testing services for adult patients, with most test referrals received from reproductive endocrinology and infertility (REI) special-ists. Screening was offered to couples planning for pregnancy or in the early stages of pregnancy. All participating couples consisted of one male and one female individual presenting as reproductive partners.”

  • Comment 3: The authors should also describe the data sources if they include information beyond what is available from the laboratory database. 

Thank you for this comment. All primary data were obtained from the laboratory’s internal clinical database. We have clarified in the supplementary material the date on which ClinVar (the only external data source that is referenced) was accessed.

  • Comment 4: The authors say that they excluded genes associated solely with mild conditions. How was this done? How was it determined that a condition is "mild"? Why were genes for mild conditions on the CarrierSeq panel?

Thank you for this comment. The exclusion of genes associated solely with mild conditions was based on clinical judgment by the first author, a genetic pathologist, and aligned with commonly accepted criteria referenced in the Introduction (page 2, from line 50): “Numerous criteria have been proposed for choosing genetic conditions for screening, such as well-defined phenotypes, early-onset disease, shortened lifespan, severe physical or in-tellectual disability, disabling malformations, significant impact on quality of life, need for medical or surgical interventions, and high population carrier frequencies [3,15–17] De-spite such selection criteria, variability in the number and types of genes screened persists among carrier screening providers and in research studies [18]”

Unfortunately, we no longer have access to the list of genes that were excluded, as they were filtered out prior to test configuration. As noted in the Introduction, considerable variability in gene panels (across clinical test providers and research studies) is inherent to carrier screening.

The CarrierSeq panel used in this study is a commercial product developed by Thermo Fisher. While the panel includes a broad range of genes, the exact rationale for their inclusion is proprietary and not publicly available to us.

  • Comment 5: The technical details about clinical validation of the panel are well written and add to the validity of this study. 

Thank you for this comment.

  • Comment 6: The authors should explain how the "core" genes were selected and why individual carrier status was disclosed for these genes. 

Thank you for this comment. The core genes were selected based on clinical and policy considerations. The three-gene carrier screen for CFTR, SMN1, and FMR1 is reimbursed by the Australian government, and as such, reporting individual carrier status for these genes is a requirement for reimbursement.

HBA1, HBA2 (alpha globin) and HBB (beta globin) were included in the core gene list due to their high carrier frequencies in several non-European populations, which may form a not insignificant portion of the patients referred to the laboratory.

We have added the following explanation to Materials and Methods (page 3, from line 132): “The core genes were selected either because Australian government reimbursement for testing is contingent on reporting carrier status in these genes (CFTR, SMN1, FMR1), or due to their known high carrier prevalence in multiple global populations (HBA1, HBA2, HBB).”

  • Comment 7: The authors should clarify the genes targeted for disclosure as secondary findings and the process that led to this policy. They state that disclosure is restricted to heterozygous variants in four genes and "potentially biallelic variants in other genes". This is markedly different than the ACMG recommendations. Does this represent a national consensus? a local consensus?

Thank you for this thoughtful comment. Among the four genes initially targeted for potential secondary finding disclosure, only three (LDLR, ATM, TTN) had reportable findings in our cohort. As outlined in the manuscript on page 7, from line 306: “Autosomal dominant findings were limited to variants in LDLR, TTN, and ATM. Alt-hough LDLR and TTN genes are included in ACMG secondary finding guidelines, the ATM gene is not included and is classified as a moderate-risk allele for breast cancer. However, both NCCN (National Comprehensive Cancer Network) and eviQ recommend annual mammography screening from age 40 for female ATM carriers, with modifications to surveillance based on individual risk assessments using validated models such as CanRisk, [29]”

Regarding biallelic variants in genes not specifically targeted for secondary finding disclosure, there is currently no national or international consensus on whether and how such findings should be reported. As highlighted in our introduction, this remains an emerging area with limited guidance. Our study did not include clinical follow-up and therefore cannot assess the clinical outcomes or utility of disclosing these findings. Nevertheless, our data contribute to the growing body of evidence on the frequency of such findings in a largely asymptomatic population. We hope this contributes to future efforts to establish best practices and consensus in this area.

  • Comment 8: It is unclear how "high" vs "low/medium" impact are defined.  In the methods, low/medium impact seems to be related to the likelihood of a clinical impact. Is this based on penetrance? Does it take into consideration variable expressivity, type of impact, actionability, or any other factor that could influence clinical impact? The authors should describe their definitions and add references where the definitions could be found, if available. They should also describe how the impact is assessed for each variant/genotype (e.g. who/how many team members assess a variant? how is consensus reached?)

Thank you for this comment. Based on this comment, the process for determining clinical impact has been further clarified in Materials and Methods (page 4, from line 149):

“The severity of at-risk conditions was classified based on the criteria established by Lazarin et al. [17]. For conditions with a known heterogeneous phenotypic spectrum, the most severe untreated disease presentation was used as the reference point for assessment. Each condition was categorised as either “severe/profound” or “mild/moderate” based on this framework. Each couple’s risk status was then further classified as either “high” or “low/medium” clinical impact, considering the specific variants identified in the couple. All findings involving mild/moderate conditions were automatically classified as low/medium clinical impact. In contrast, findings related to severe/profound conditions were individually reviewed by a genetic pathologist, who assessed clinical impact based on factors such as penetrance, variable expressivity, and published evidence regarding specific genotype combinations in the literature and online databases.”

We acknowledge that this determination by a single individual is a weakness of the study and have included this in the limitations section (page 8, line 338):

“Second, the classification of clinical impact, which informed the adjusted at-risk couple rate, was performed by a single pathologist. While this introduces some subjectivity, the classifications were based on current medical knowledge, and the key sources used for these evaluations have been made available in the Supplementary Materials.”

  • Comment 9: Figure 1 should be referenced in the results, not the methods. 

Thank you for pointing this out. We agree with the suggestion and have moved both Figure 1 and its first in-text reference to the Results section.

  • Comment 10: The numbers in figure 1 are a little confusing:
    • Comment 10a: The boxes that list how many core genes vs other genes are just a description of the box to their left, not an additional step in the process. They should be represented differently. 
    • Comment 10b: if including the step "excluding low/moderate impact results", the authors should also include a step described as "excluding mild/moderate conditions"
    • Comment 10c: the tables use the terminology "low/medium" impact, whereas the figure uses the term "low/moderate" impact. The same terminology should be used throughout the manuscript. 
    • Comment 10d: if including all the different reasons to consider a condition to be "low/medium impact", it would be more interesting to know how many conditions were excluded for each of these reasons. 

Comment 10a: Thank you for this helpful observation. We agree that the original layout may have suggested an additional analytical step. To clarify this, we have revised the visual presentation by replacing the arrows with plain lines to indicate that these are descriptive annotations rather than sequential steps.

Comment 10b: Thank you for this suggestion. We have now updated Figure 1 to include separate boxes representing the exclusion of mild/moderate conditions and low/medium clinical impact results, along with the corresponding case numbers, to improve clarity and better represent the filtering process.

Comment 10c: Thank you for noting this inconsistency. We have reviewed the manuscript and figures and have now standardised the terminology to “low/medium clinical impact” throughout.

Comment 10d: This is an insightful suggestion. However, in many cases, multiple factors contributed to a classification of “low/medium impact,” making it difficult to assign cases to a single exclusion criterion without oversimplifying. Instead, we have provided illustrative examples for each classification category in the Figure legend to enhance interpretability without introducing misleading quantification.

  • Comment 11: I suggest moving the sentence "Among FMR1 carriers, the majority (73%) had 55–64 repeats." right after the sentence about alpha-thalassemia carriers, because it describes carriers, before moving to the description of at-risk couples for the core conditions.

Thank you for this helpful suggestion. We have implemented this change and agree that it improves logical flow.

  • Comment 12: The results in the paragraph reporting carriers for core genes are confusing because they report absolute numbers, ratios (1 in X) and percentages. The percentages are about carriers of a specific condition, but don't all relate back to the total number of carriers for that condition. For example, only a percentage is given for FMR1 carriers, whereas absolute numbers and percentages are given for alpha-thalassemia carriers. 

Thank you for this feedback. In response, we have kept the absolute numbers and ratios (1 in X) for the frequency of carriers in each gene, as carrier frequencies in the cohort are the most clinically pertinent findings in this section.

For HBA1/HBA2 and FMR1, we have included absolute numbers and percentages for different variant types, as these distinctions provide important clinical context. Specifically, that most alpha thalassemia carriers had a single-gene deletion, and most FMR1 carriers had low premutation alleles.

We have removed the percentage of individuals found to be a carrier of one or more core genes, as we determined this aggregate figure did not contribute meaningful insight.

  • Comment 13: Supplementary table S1 is not very helpful: some diseases classified as severe/profound are classified as having a low/medium impact and it isn't clear how the severity of the predicted phenotype has led to a downgrade of the impact (e.g. classic congenital adrenal hyperplasia is classified as a "severe/profound disease" but with a "low/medium" impact.

Thank you for this comment. The congenital adrenal hyperplasia example is a particularly important one, and we have now included an explanation in the Supplementary Materials detailing the rationale for its reclassification:

“Although classic congenital adrenal hyperplasia (CAH) can be predicted in offspring who are homozygous for p.(Gln319*), approximately 84% of p.(Gln319*) alleles are known to be in cis with a whole-gene duplication (PMID: 19773403). This significantly reduces the likelihood that one or both partners are true CYP21A2 carriers, and therefore the risk of classic CAH in their offspring is likely to be low.”

In addition, we have reviewed all other instances where conditions typically classified as severe or profound were reassigned a low or medium clinical impact rating, and ensured that justifications for each of these reclassifications are clearly documented.

  • Comment 14: About secondary findings, they seem to have included variants of unknown significance and of conflicting classification (table S3). This is at odds with the ACMG guidelines. How do the authors justify reporting such findings?

Thank you for this comment. The column showing ClinVar classifications in Table S3 is included for informational purposes only and does not reflect the final classification assigned by our laboratory. All variants that were included in clinical reports, whether carrier states or secondary findings, were independently assessed and classified by the laboratory as pathogenic or likely pathogenic. No variants classified by our laboratory as being of uncertain significance were reported.

To clarify this point, we have added the following sentence to the Materials and Methods section (page 3, from line 119): “All reported variants, whether identified as carrier states or secondary findings, were clas-sified by the laboratory as pathogenic or likely pathogenic. Variants of uncertain signifi-cance were not reported.”

  • Comment 15: For secondary findings for autosomal recessive phenotypes, the pathogenicity classification of variants is not reported. 

Thank you for this comment. As noted above, we can confirm that we only included variants assessed by our laboratory as pathogenic or likely pathogenic.

For the table referencing secondary findings in genes associated with autosomal recessive conditions, we did not include ClinVar classifications due to table space constraints, as each case involved two variants.

  • Comment 16: The authors argue that broader carrier screening provides a more comprehensive identification of couples at risk for AR or X-linked conditions and that the currently funded public option in Australia may lead to unequal access to broader screening.  They do not address the added costs and resources needed for broader screening, nor the social acceptability of broader screening. 

Thank you for this important observation. We agree that considerations such as the additional costs, resource implications, and social acceptability of broader carrier screening are critical to the broader implementation discussion. However, these factors were outside the scope of our current study and were not addressed in our analysis. We acknowledge the importance of these issues and agree they warrant further research and policy discussion.

  • Comment 17: The author rightly acknowledge that their study population is enriched in women with FMR1 carrier status. 

Thank you for this comment.

  • Comment 18: Authors discuss the percentage of at-risk couples identified with different strategies. It seems that part of their argument is that the test may target a broader list of genes/conditions but the number of at-risk couples identified can remain manageable if only those with variants in genes associated with severe/profound conditions and predicted high impact are reported. Is that what the authors are arguing? This could be made clearer in the discussion. 

Thank you for this thoughtful comment. Our argument is closely aligned with what is stated here, with an important clarification. Rather than suggesting that the number of at-risk couples remains manageable, our key point is that carrier screening outcomes should be interpreted and reported in the context of predicted disease severity and variant-specific clinical impact, as these factors directly influence reproductive decision-making.

When these post-test considerations are applied, the proportion of couples for whom the results are actionable before or during pregnancy is lower than the unadjusted at-risk couple rate often reported. This has important implications for both patients and policy makers.

As carrier screening expands in scope and moves toward public funding or reimbursement, the reported rate of at-risk couples will become a key metric for assessing clinical utility and cost-effectiveness. It is therefore critical that these figures reflect clinically meaningful risk, rather than overstating potential benefits on a population-wide basis.

A related point is that consideration of variant combinations within couples is rarely performed at the laboratory level, despite its relevance to predicting disease severity in potential offspring. Given its impact on clinical interpretation and reproductive planning, we argue that this should become standard practice wherever couple-based reporting is available.

These points have been included in the following paragraphs in the Discussion (page 7, lines 283-296):

“These findings have important implications for policy makers. As carrier screening expands in scope and increasingly moves toward public funding or reimbursement, the reported rate of at-risk couples will become a key metric for assessing clinical utility and cost-effectiveness. It is therefore important that these figures reflect clinically meaningful risk, rather than overstating the potential benefits on a population-wide basis.

This study also reinforces existing calls to integrate variant combination analysis into carrier screening for autosomal recessive conditions [12,18–20] Prior recommendations have even proposed the creation of an international database of autosomal recessive vari-ant combinations and their phenotypic outcomes [19] which could enhance both carrier screening and genomic newborn screening by improving phenotype prediction. However, this type of analysis is currently rarely performed at the laboratory level. Our findings suggest that variant combination analysis, given its impact on clinical interpretation and reproductive planning, should become standard laboratory practice wherever couple-based reporting is available.”

  • Comment 19: Authors should discuss the challenges of predicting clinical impact on the basis of specific variants.

Thank you for this important comment. As noted in our response to Comment 8, we have addressed this point by explicitly including it as a limitation of the study.

  • Comment 20: The inclusion of secondary findings in the context of carrier screening could be discussed in greater detail, especially if not currently done clinically. 

Thank you for this thoughtful comment. In response, we have revised the relevant section of the Discussion (Section 4.3, page 7) to clarify what we defined as a secondary finding, refine the wording for clarity and precision, and added commentary on areas where further research is needed to inform clinical practice.

  • Comment 21: Avoid the formulation "real world couples". Even in research studies, participating couples are real world couples. The emphasis should be on the testing context, not the couple. For example, "this study has several strengths, including the comprehenisve detection of genetic variants in couples testing in real-world settings"

Thank you for this comment. We agree that the emphasis should be on the testing context rather than the participants. All references to “real-world” have been revised to “routine clinical setting” to more accurately reflect the nature of the testing environment.

Round 2

Reviewer 2 Report

Comments and Suggestions for Authors

I appreciate the authors' thoughtful revision to their manuscript. I have only one remaining issue after reviewing their comments and edits. Given that the authors “recognise the importance of understanding the genetic ancestry composition of the cohort” but cannot assess ancestry of the cohort, this should be acknowledged as a limitation in the discussion.

Author Response

Thank you for your thoughtful feedback and for taking the time to review our revisions.

We have incorporated your suggestion into the Discussion (page 8, lines 338–340) with the following sentence:
“Second, the lack of self-reported or genotype-based ancestry information among study participants limits our ability to assess representation across the broader population.”

We are grateful for your insights, which have meaningfully improved the clarity and completeness of the manuscript.

Reviewer 3 Report

Comments and Suggestions for Authors

My comment is still the same

This paper is just an advertisement and no scientific work

Author Response

Thank you for your comment. We respectfully disagree with your assessment, but appreciate you taking the time to review the work.

Reviewer 4 Report

Comments and Suggestions for Authors

The authors have provided thoughtful answers to the comments, and made appropriate modifications to the manuscript. 

Author Response

Thank you for your kind review. We appreciate your thoughtful feedback and are glad the revisions addressed the comments appropriately. We’re grateful for your time and consideration in helping us improve the manuscript.